# CBD and THC in Special Populations: Pharmacokinetics and Drug–Drug Interactions

**DOI:** 10.3390/pharmaceutics16040484

**Published:** 2024-04-01

**Authors:** Lixuan Qian, Jessica L. Beers, Klarissa D. Jackson, Zhu Zhou

**Affiliations:** 1Department of Chemistry, York College, City University of New York, Jamaica, NY 11451, USA; lqian@york.cuny.edu; 2Division of Pharmacotherapy and Experimental Therapeutics, Eshelman School of Pharmacy, University of North Carolina at Chapel Hill, Chapel Hill, NC 27599, USAklarissa.jackson@unc.edu (K.D.J.)

**Keywords:** cannabidiol, delta-9-tetrahydrocannabinol, drug–drug interaction, special population

## Abstract

Cannabinoid use has surged in the past decade, with a growing interest in expanding cannabidiol (CBD) and delta-9-tetrahydrocannabinol (THC) applications into special populations. Consequently, the increased use of CBD and THC raises the risk of drug–drug interactions (DDIs). Nevertheless, DDIs for cannabinoids, especially in special populations, remain inadequately investigated. While some clinical trials have explored DDIs between therapeutic drugs like antiepileptic drugs and CBD/THC, more potential interactions remain to be examined. This review summarizes the published studies on CBD and THC–drug interactions, outlines the mechanisms involved, discusses the physiological considerations in pharmacokinetics (PK) and DDI studies in special populations (including pregnant and lactating women, pediatrics, older adults, patients with hepatic or renal impairments, and others), and presents modeling approaches that can describe the DDIs associated with CBD and THC in special populations. The PK of CBD and THC in special populations remain poorly characterized, with limited studies investigating DDIs involving CBD/THC in these populations. Therefore, it is critical to evaluate potential DDIs between CBD/THC and medications that are commonly used in special populations. Modeling approaches can aid in understanding these interactions.

## 1. Introduction

### 1.1. Brief Overview of CBD and THC

“Cannabis” refers to several plants of the genus *Cannabis* (family Cannabaceae), primarily *C. sativa*, which contains > 500 constituents, including > 100 cannabinoids [1,2,3]. The two most extensively studied cannabinoids are the non-psychotoxic cannabidiol (CBD) and the psychoactive delta-9-tetrahydrocannabinol (THC) [4]. The psychoactive effects of cannabinoids are mainly mediated by cannabinoid receptor 1 (CB1) receptors [5], and THC is the agonist of both CB1 and cannabinoid receptor 2 (CB2) [6]. CBD is considered non-psychotoxic, as it does not bind to the cerebral CB1 receptor [7,8]. Instead, it exerts a negative allosteric modulation on CB1 [8,9]. CBD is a low-affinity agonist of the CB2 receptor [10]. Since the first approval of CBD and the synthetic version of THC by the FDA, there has been a growing interest in CBD and THC clinical trials to treat epilepsy, pain, depression, anxiety, psychosis, neurodegenerative disease, gastrointestinal diseases, and addiction [10,11]. CBD was the top-selling herbal supplement ingredient in the US natural channel from 2018 to 2020, increasing by >450% from 2017 [12]. Purified CBD (brand name: Epidiolex^®^) has been approved in the United States and European Union for the treatment of refractory epileptic seizures in Dravet and Lennox–Gastaut patients, as well as seizures associated with tuberous sclerosis complex [13,14,15]. Meanwhile, cellular and rodent studies also suggest that CBD has therapeutic potential as a cardioprotective and anti-inflammatory agent [16,17,18]. The mean THC content in cannabis-derived products was reported to increase 2-fold from 2009 to 2017 [19]. To date, the U.S. Food and Drug Administration (FDA) has approved two drugs containing a synthetic version of THC (dronabinol) and one drug that is a synthetic analog of THC (nabilone). Both are indicated for nausea and vomiting associated with cancer chemotherapy; dronabinol is also indicated for anorexia associated with weight loss in patients with acquired immunodeficiency syndrome (AIDS) [20,21]. The 1:1 mixture of purified THC and CBD as an oromucosal spray (brand name: Sativex^®^) has been approved in the United Kingdom and other countries for the treatment of multiple sclerosis [22,23]. CBD/THC on the market and in development have various routes of administration, including oral, inhaled/vaporized, sublingual, and as an oromucosal spray [10].

### 1.2. Importance of Understanding Pharmacokinetics and Drug Interactions

Both therapeutic effects and adverse effects increase with exposure to CBD [24,25]. The proportion of patients experiencing transaminase elevations that are greater than three times the upper limit of the reference range in phase III trials increased from 6.8% at a CBD dose of 10 mg/kg/day to 27.7% at a CBD dose of 20 mg/kg/day, compared to 0.0% for the placebo [25]. THC doses above 10 mg are reported to be associated with an increased risk of addiction and THC-related psychosis [26,27]. For example, in a study in the UK, higher-potency cannabis users reported being four times more likely to have cannabis use problems than lower-potency cannabis users [28]. Pichini et al. suggested that smoking four cannabis cigarettes is associated with more severe drowsiness than smoking one cigarette [29]. Therefore, it is crucial to understand the pharmacokinetics (PK) of CBD and THC to determine the appropriate dose, maximize the treatment effect, and minimize the risk of adverse effects and addiction. However, the PK of CBD and THC exhibits great inter-individual variability in absorption, distribution, and metabolic clearance [4,30,31].

CBD is classified as a class II Biopharmaceutical Classification System (BCS) substance [32]. Its low water solubility and high passive permeability lead to poor and erratic oral absorption. CBD’s estimated absolute oral bioavailability is 6% with an empty stomach [33], increasing to 14–23% with food [34]. CBD’s relatively low oral bioavailability can be attributed to incomplete gastrointestinal absorption and extensive first-pass metabolism [33]. Distribution and elimination also play crucial roles in CBD’s highly variable PK. CBD’s high lipophilicity leads to a large volume of distribution and substantial accumulation in the adipose tissue. The volume of distribution (Vd) of CBD in adults ranged from 2.5 to 10 L/kg [35]. CBD concentrations are sensitive to body weight and adiposity. CBD’s metabolism involves both cytochromes P450 (CYP) and UDP-glucuronosyltransferases (UGT) [36,37,38]. Polymorphisms in these metabolic enzymes may introduce variability in PK, which will be discussed in Section 2.3. In plasma, the terminal elimination half-life (t_1/2_) of CBD varied from 24 to 61 h [35]. Most CBD and its metabolites are excreted through feces, and a limited part is excreted through urine [4].

Similar to CBD, THC is classified as a BCS class II drug [32], exhibiting limited and variable oral absorption, with an oral bioavailability of 4 to 20% [39]. THC is also widely distributed in fat and has a large Vd, about 10 L/kg [39]. Unlike CBD, THC undergoes metabolism solely through CYP enzymes, including CYP2C9, 2C19, and 3A4, although THC metabolites are glucuronidated [38,40,41]. Changes in the activity of these CYP enzymes could impact the PK of THC. Furthermore, sex differences in the PK of THC have been observed in both animal and clinical trials [42,43]. THC has a long terminal t_1/2_ of about 22 h and is excreted in feces and urine as glucuronidated metabolites [4].

As CYP enzymes contribute to the metabolism of both CBD and THC, medications affecting these enzymes could potentially lead to drug–drug interactions (DDIs). Therefore, a comprehensive understanding of drug interactions is crucial for optimizing therapeutic outcomes. This knowledge guides dosage adjustments and helps mitigate potential risks associated with variable drug responses.

### 1.3. Significance of Studying Special Populations

Another challenge in determining CBD and/or THC doses lies in the fact that the current PK knowledge of CBD and THC is derived from studies conducted on healthy adults. The impact of different factors, including but not limited to aging, obesity, and disease conditions, has not been fully assessed. For example, due to the highly lipophilic nature of CBD/THC, the distribution may be different among the pediatric population, older adults, and individuals with obesity. While prior publications have reviewed the PK of CBD and THC in healthy adults [4,44,45,46,47], there is a significant gap in current understanding when it comes to special populations. This review aims to address this gap by focusing on the PK and potential DDIs of CBD and THC in special populations.

## 2. CBD/THC PK-Based Drug Interactions

### 2.1. Overview of Drug Interactions Involving CBD and THC

With the increasing use of recreational and medical cannabis, drug interactions involving CBD and THC have raised concerns. While Sativex^®^ was approved in the UK in 2011, widespread attention to the issue of DDIs involving CBD and/or THC began with the approval of Epidiolex^®^, which is used in the treatment for certain seizures in children and adults. It is, therefore, not surprising that most DDI studies have focused on investigating the interaction between CBD/THC and antiepileptic drugs in pediatric, healthy adults, or young seizure patients [48,49,50,51,52]. In pediatrics, the DDI between CBD and antiepileptic drugs, including stiripentol [50], valproate [50,51], and clobazam [51,52,53], was tested and validated. In young adults, antiepileptic drugs (including stiripentol [48,50], valproate [48,50,51], tiagabine [54], and clobazam [48,49,51]), as well as warfarin [55,56,57,58,59], clozapine [60], buprenorphine [61], tacrolimus [62], sirolimus [63], everolimus [63,64], caffeine [65], citalopram [66], and theophylline [67,68], have been reported as victims of CBD/THC according to clinical trials and/or case reports. Conflicting results were observed in the DDI effect of CBD and THC on phenobarbital [51,69]. Also, the effect of typical DDI perpetrators, including ketoconazole and rifampicin, on CBD and THC was assessed in young adults [48,70].

In addition to healthy adults, some studies have reported potential DDIs between CBD/THC and certain medications in special populations. For instance, interactions were noted with medications like clopidogrel that was co-administered with aspirin (76-year-old male) in older adults [71]. A recently published systematic review concluded that a DDI between warfarin and CBD/THC is probable, although the available evidence was limited to only seven published case reports [72]. In a separate trial involving human immunodeficiency virus (HIV)-infected patients, no interaction was reported between THC and indinavir or nelfinavir [73]. Additionally, investigations into potential DDIs between CBD/THC and addictive substances, such as methadone [74], fentanyl [75], morphine [76], and cocaine, were conducted [77]. Vierke et al. also characterized the DDI between buprenorphine and CBD/THC [61]. Additional studies on drug interactions involving CBD and THC in special populations will be discussed in the subsequent sections.

### 2.2. Mechanisms of CBD and THC Interactions with Other Drugs

The mechanisms of DDIs involving CBD/THC are primarily PK-based. Major phase I metabolizing enzymes play key roles, particularly CYP3A4, CYP2C9, and CYP2C19. As a result, DDIs can occur when an inhibitor and/or a substrate of these CYP enzymes is co-administered with CBD/THC. The major metabolizing enzymes and metabolites of CBD are summarized in Table 1. CBD is metabolized mainly to 7-OH-CBD by CYP2C9 and CYP2C19, while CYP3A4 is crucial for other hydroxylation products [37,78]. 7-OH-CBD is further metabolized to the inactive 7-COOH-CBD. The primary metabolizing enzyme for this step remains unidentified, although recent evidence suggests that both CYP and aldehyde dehydrogenase (ALDH) enzymes are involved [79]. 7-OH-CBD and 7-COOH-CBD can undergo glucuronidation in phase II metabolism, but the specific UGTs that are involved require further study. Inducers and inhibitors of the metabolic enzymes (as listed in Table 1) have the potential to interact with CBD and its metabolites. Studies present inconsistent conclusions regarding whether CBD is a substrate for transporters. In silico studies suggest that CBD acts as a P-glycoprotein (P-gp, also known as multidrug resistance protein 1, MDR1, or ATP-binding cassette subfamily B member 1, ABCB1) substrate [80], while animal studies propose that CBD is not a substrate for P-gp or the breast cancer resistance protein (BCRP) [81]. In vitro, the major metabolite 7-COOH-CBD is a substrate for P-gp and an inhibitor of BCRP and the bile salt export pump [14]. A study from Holland et al. found that CBD strongly inhibited ATP-binding cassette subfamily C member 1 (ABCC1, also known as multiple-drug-resistant transporters 1, MRP1) in ovarian carcinoma cells overexpressing MRP1 [82]. However, to the authors’ knowledge, this has not been investigated in other cell types.

CBD carries the potential for numerous drug-metabolizing enzyme-mediated DDIs, and multiple instances of these interactions have been reported in the literature [51,85,87]. Enzymes that are impacted by CBD and the related parameters from in vitro studies are listed in Table 1. Current Epidiolex^®^ prescribing information advises dosage adjustment of CBD with strong inducers of CYP3A4 or CYP2C19, as well as monitoring patients receiving concomitant substrates of CYP1A2, CYP2C8, CYP2C19, UGT1A9, or UGT2B7 for adverse effects associated with an inhibited metabolism of these substrates [14]. For transporters, Alsherbiny et al. noted in their review that the P-gp expression is sensitive to the duration of CBD exposure, being down-regulated in chronic exposure and up-regulated in short-term exposure [88]. 

THC is the substrate of CYPs in the phase I metabolism [41,83,84,89]. The major metabolizing enzymes and metabolites of THC are summarized in Table 1. A portion of the active main metabolite, 11-OH-THC, undergoes glucuronidation by UGT1A1 and UGT1A3 in the phase II metabolism [41,79,83,84]. The metabolite 11-COOH-THC is a substrate of the UGT1A family, primarily UGT1A9 and UGT1A10, in phase II metabolism [38]. Similar to CBD, there are conflicting results regarding whether THC is a substrate of transporters. Animal studies suggested that THC is the substrate of P-gp and BCRP [90,91]. In contrast, Chen et al. concluded from a Madin–Darby Canine Kidney cell study and an animal study that THC and most of its metabolites are not substrates of P-gp and BCRP, except for 11-COOH-THC, which is a poor substrate of BCRP [92,93].

Despite being the substrate of enzymes such as CYPs and UGTs, THC and its metabolites are also inhibitors of CYPs [65,85], MRP1 [82], and carboxylesterase 1 (CES1) in vitro [94]. According to another in vitro study, 11-COOH-THC is an inhibitor of BCRP [92]. Substrates of such enzymes are potential victims in DDIs involving THC. Enzymes that are impacted by THC and the related parameters from in vitro studies are listed in Table 1. Brzozowska et al. proposed that THC might increase the P-gp expression at the blood–brain barrier, leading to a decrease in the concentration of risperidone in the brain [95].

Many clinical trials have demonstrated DDIs with CBD and THC as perpetrators. Vierke et al. found that the CYP3A4 substrates buprenorphine and norbuprenorphine’s concentration were 2.7-fold and 1.4-fold higher in cannabis users [61]. When receiving a daily CBD dose of 2000–2900 mg, a 3-fold increase in normalized tacrolimus (CYP3A substrate) concentrations was reported by Leino et al. [62]. According to Jusko et al., when volunteers were exposed to cannabis, the clearance of theophylline (a CYP1A2 substrate) increased from 51.8 mL/h/kg to 73.3 mL/h/kg [68]. Thai et al. suggested that when another CYP1A2 substrate, caffeine, was co-administered with CBD (750 mg twice a day (b.i.d.)), the area under the curve (AUC) of caffeine increased by 95% [65]. One of the most studied CBD-induced DDIs is with clobazam, an antiepileptic drug that is often co-prescribed with CBD in patients with Lennox–Gastaut syndrome [48]. Clobazam is metabolized to the active metabolite *N*-desmethylclobazam by CYP3A4, and further hydroxylation to the inactive metabolite 4′-hydroxydesmethylclobazam is primarily mediated by CYP2C19 [96]. The inhibition of these P450 enzymes by CBD has been shown to significantly increase the exposure to *N*-desmethylclobazam, which has been demonstrated to increase the risk of adverse effects, including sedation, ataxia, irritability, tremor, urinary retention, and loss of appetite [48,51,53]. In a clinical trial involving ten male and ten female patients with epilepsy, CBD increased the AUC of the dosing interval and the C_max_ of *N*-desmethylclobazam by 2.6-fold and 2.2-fold, respectively, when administered at a dose of 20 mg/kg/day [49]. As mentioned previously, there are also multiple case reports suggesting that cannabis can decrease the anticoagulant effect of warfarin, a CYP2C9 substrate [56,57,58,59,97,98]. Of note, CBD can act as a perpetrator of DDIs when combined with THC by inhibiting the metabolism of THC and 11-OH-THC. A recent clinical DDI study with eleven males and seven females found that orally administered CBD increased the oral exposure to THC and its metabolites 11-OH-THC and 11-COOH-THC when co-administered with a CYP probe drug cocktail [99]. Furthermore, patients receiving both CBD and THC experienced increased self-reported anxiety, sedation, and memory difficulties, as well as increased impairment of psychomotor and cognitive function compared to THC alone [99]. These clinical trials suggest that when co-administered with CYP2C9, 2C19, and 3A4 substrates, CBD and THC can lead to clinically significant DDIs (change in AUC ≥ 25% for inhibition, according to FDA guidance [100]) when acting as perpetrators. However, some clinical trials showed that THC did not inhibit the metabolism of some CYP2C19, 2C9, and 3A4 substrates as expected. Substrates of the enzymes that THC can inhibit, including the CYP3A4 and CYP2C19 substrate nelfinavir [73]; the CYP3A4 substrates indinavir [73], irinotecan [101], docetaxel [101], and doxorubicin [102]; and cyclophosphamide [102], a substrate of CYP2C9, 2C19, and 3A4, have been reported not to interact with THC. For example, the metabolism of nelfinavir (a substrate of CYP2C19 and 3A4) and indinavir (a substrate of CYP3A4) was expected to be inhibited by THC, as THC was reported as a CYP2C19 and 3A4 inhibitor in vitro, but a clinical trial reporting the statistically non-significant steady-state AUC decreases of 10.2% (*p* = 0.15) and 14.5% (*p* = 0.074), respectively, disproved this hypothesis [73]. In a clinical trial involving 14 male and 10 female cancer patients, there was no significant increase in the AUC with cannabis administered as an herbal tea for irinotecan (1.04-fold) or docetaxel (1.11-fold), even though irinotecan and docetaxel are mainly metabolized via CYP3A [101]. Similarly, Riggs et al. suggested that when used with THC, no DDI effect on the exposure to doxorubicin (a CYP3A4 substrate) and cyclophosphamide (a CYP2C9, 2C19, and 3A4 substrate) was observed. CBD is suggested to inhibit CES1 in vitro, but methylphenidate, a substrate of CES1, was reported to have no interaction with CBD [103]. More studies are necessary to investigate the mechanisms involved in these interactions between CBD/THC and drugs.

In addition to enzyme-mediated DDIs, several other factors might impact the PK of CBD and THC, as listed in Table 1. Due to its high lipophilicity, THC can accumulate in adipose tissue. Drugs that alter the total fat mass, fat distribution, or fat composition, such as glucocorticoids and diet pills, may influence the distribution of THC. Furthermore, 95–99% of plasma THC is bound to plasma proteins. Hence, competitive protein binding with other drugs may also change the PK of THC [104,105]. Additionally, there is controversy over whether CBD can be converted to THC in the body [106]. Some in vitro and animal studies suggested that CBD can be transformed into THC in simulated gastric fluid and rats [107,108]. However, a clinical trial designed specifically for the bioconversion of CBD to THC did not find such conversion [109].

### 2.3. Genetic Variations May Impact CBD/THC PK and Drug Interactions

Some DDIs that have been identified in vitro studies are deemed not clinically significant when validated in vivo. The differences between an in vitro prediction and the in vivo results may be explained by the drug–drug–gene interaction (DDGI) [110]. A DDGI can boost the DDI by affecting the metabolism pathways. However, a DDGI can also lead to a contrary effect with the DDI and diminish the DDI. The metabolism phenotype of the enzymes can lead to different types of DDGIs when there is an inhibitor or inducer [110]. Bahar et al. summarized the DDGIs on CYP2C9, CYP2C19, and CYP2D6. They suggested that the normal metabolizer phenotype leads to the greatest distortion of the DDIs, followed by the intermediate metabolizer and the least in the poor metabolizer [111]. CYP2C9, CYP2C19, and CYP2D6 are involved in the metabolism of 12.8%, 6.8%, and 20% of the majority of clinical drugs, respectively [112]. Their polymorphisms are the most frequently occurring variants in the phase I metabolism of drugs [113]. Bland et al. investigated the effects of polymorphism on CYP2C9 and found that the clearance for recombinantly expressed *CYP2C9*2* and *CYP2C9*3* were only 30% of that of the *CYP2C9*1* (wild-type) [89]. Wolowich et al. investigated the effect of CYP2C9 polymorphism using physiologically based pharmacokinetic (PBPK) modeling and identified significant differences (*p* = 0.02) in the THC clearance between *CYP2C9*3* homozygotes and other phenotypes. The total clearance of patients with CYP2C9 (homozygous/wild-type) was 41.9% of CYP2C9 (heterozygous/heterozygous) [114]. Additionally, Wolowich et al. also found that subjects who are homozygous for *CYP2C9*3* have a 74% decrease in their THC-COOH exposure [114]. When predicting DDIs involving CYP2C9 using only wild-type CYP2C9, the contribution of the enzyme may be overestimated, leading to biased results. Given that CBD and THC can be metabolized by multiple CYPs, the importance of the second pathway will increase when the primary enzyme is reduced by a polymorphism. A more significant change in PK may be observed when both the primary and second pathways are influenced by DDIs, which may occur in polypharmacy.

## 3. CBD/THC Studies in Special Populations

In this section, studies on the effects of CBD and THC in special populations, including pregnant and lactating women, pediatrics, older adults, patients with hepatic or renal impairment, cancer patients, patients with neurological disorders, and others, are reviewed. We searched three databases (The Drug Interaction Database (DIDB^®^), PubMed, and Google Scholar) from inception until March 2024 to identify clinical trials on CBD/THC in special populations. The search terms included “CBD or THC” combined with the specific populations of interest. Trials that did not provide detailed PK parameters of CBD or THC in special populations were excluded from the tables but were still described in subsequent sections. Table 2 and Table 3 summarize the PK parameters of CBD and THC in clinical trials for special populations, respectively. PK parameters in healthy volunteers were not included as a reference due to variations in administration routes and a high PK variability.

### 3.1. Pregnancy and Lactation

#### 3.1.1. Overview of Research on CBD/THC Use during Pregnancy and Lactation

The prevalence of self-reported past-month cannabis use among pregnant women in the US rose from 3.4% in 2003 to 7.0% in 2017 [127]. A study in California also showed that 19% of pregnant women aged 18 to 24 years used cannabis in 2016 [128]. Despite these trends, a limited number of studies have investigated the PK of CBD and THC and their metabolites in pregnant or lactating women. And there is a lack of cannabinoid plasma concentration–time data for pregnant or lactating women [129]. The FDA mandates postmarketing pregnancy outcomes research on Epidiolex^®^, and future research may rely on any forthcoming data from pregnant patients taking CBD who are enrolled in registries [130]. Metz et al. tested the serum, urine, and umbilical cord concentration of 11-COOH-THC in pregnant women from the gestational age of 16 weeks to delivery. Although the serum concentrations of 11-COOH-THC were measured, the trial failed to determine the THC concentrations [131]. Westin et al. reported that 11-COOH-THC was still detectable at week 14 after the last THC administration in one pregnant woman [132]. Baker et al. detected the THC levels in the breast milk of eight lactating women, suggesting that a mean of 2.5% of the THC dose was transferred into breast milk [133]. A 2018 study noted observable differences among THC, 11-OH-THC, and CBD in the breast milk of lactating women who used cannabis. THC was detected in the breast milk of 63% of the mothers, whereas only 9.0% showed detectable levels for either 11-OH-THC or CBD [134]. This pattern was also observed in a recent study, where 13 out of 14 breast milk samples showed THC levels, while detectable levels of 11-OH-THC and CBD were found in 3 and 1 sample, respectively. Additionally, this research reported that cannabis use also alters the lactose- and soluble IgA levels in the milk. Other CBD and THC studies in pregnant women and their infants focus on qualitative research on the adverse effects of CBD/THC on infants, e.g., malformations and effects on the development of male gonads [135,136]. No DDI studies involving CBD or THC in pregnant or lactation women were found. A PBPK model of THC was established by Patilea-Vrana et al. and extrapolated to pregnant women. The simulations showed no clinically significant change in the AUC_0–t_ or C_max_ of THC between pregnancy and nonpregnancy, but the AUC_0–t_ and C_max_ of 11-OH-THC decreased by up to 41% and 44%, respectively, during pregnancy compared with nonpregnancy [137]. However, this extrapolation was not validated due to a lack of clinical data. Shenkoya et al. developed and verified a paired lactation and infant THC PBPK model. When the authors simulated the inhalation of 0.045 g of THC, they found that the mean AUC and C_max_ of breastmilk were 2.2 and 3.4 times higher than that of plasma, respectively [138].

#### 3.1.2. PK Considerations in Pregnant and Lactating Women

Pregnant and lactating women undergo physiological changes that may result in altered PK of CBD or THC. However, the overall impact of pregnancy and lactation on CBD and THC exposure remains unknown. An individual’s body fat increases during pregnancy due to increased adipogenesis in the adipose tissue and adipocyte hyperplasia [139]. The fat stores in pregnant women reach their peak towards the end of the second trimester and then diminish [139,140]. Consequently, there may be changes in the volume of distribution of CBD/THC and their metabolites during pregnancy, potentially leading to variations in concentrations with gestation. Cardiovascular changes occur during pregnancy, including an increase in heart rate, a rise in plasma volume, an increase in plasma lipids, and a decrease in plasma protein levels. The increase in plasma volume may result in lower plasma CBD/THC concentrations, while the reduction in plasma protein levels may exacerbate the concentration decrease. In contrast, the free cannabinoid concentrations may increase due to the decreased plasma protein level, as CBD and THC have high protein binding (95–99% for THC) [104,105]. For elimination, Grant et al. suggested that the clearance of THC during pregnancy can increase up to 2-fold, and the THC exposure may be decreased [141]. Lastly, the cannabinoid concentration in non-blood matrices can be a good alternative. However, data interpretation could be challenging due to a lack of information [142].

#### 3.1.3. Potential Drug Interactions and Safety Concerns for this Population

The use of cannabis by pregnant women should be carefully considered, as prenatal cannabis use can be associated with potential harm to the infant, including a lower birth weight, preterm birth, neonatal intensive care unit admission, and attention and externalizing problems [143]. However, due to the high utilization rate and the unavoidable passive cannabis intake, future studies may focus on how to decrease the harm of exposure to cannabis during pregnancy. Additionally, physiologic changes during pregnancy, such as hypertension during pregnancy and gestational diabetes, may necessitate the use of medications for some pregnant women. There is a knowledge gap in DDI studies between related drugs and CBD/THC, and further study is required.

### 3.2. Pediatrics

#### 3.2.1. Overview of Studies Investigating CBD/THC in Pediatric Populations

Since the FDA approval of purified cannabidiol in the treatment of pediatric epilepsy, there has been a surge in research on the medical uses of CBD in pediatrics [144]. Most studies focus on the pharmacodynamics (PD) and the potential treatment effect of CBD and THC. In addition to the approved use for treatment-resistant epilepsy in pediatrics, research has explored the effect of CBD in oncology [145], autism spectrum disorder [146], severe behavioral problems in children with intellectual disability [147], fragile X syndrome [148], and complex motor disorder [149]. One clinical trial found that THC is a significant anti-nausea and anti-vomiting agent in the pediatric population. In some cases, THC has been shown to enhance individuals’ appetite [150]. The positive THC treatment effects were further investigated in complex motor disorders and spasticity [149,151]. However, adverse effects such as fatigue, sedation, nausea, diarrhea, appetite suppression, somnolence, and hallucinations were reported in the clinical CBD trials [152,153]. Additionally, Smid et al. suggested that THC-COOH exposure was associated with worse attention scores in early childhood [154].

PK descriptions of CBD and THC in pediatrics are limited. Wheless et al. characterized the high inter-individual variability in systemic cannabidiol exposure in pediatric patients with treatment-resistant epilepsy, and the PK parameters can be found in Table 2. They further noted that the inter-individual variability decreased during the long-term use of CBD and that CBD exposure in infants was approximately 40% of that in children or adolescents [52]. Other studies also reported a large inter-individual variability of over 80% in all PK parameters except for T_max_ in CBD exposure. They observed a disproportionate increase in CBD exposure with ascending dose over a clinically relevant dose range from 5 to 20 mg/kg/day (Table 2) [115,155]. In contrast, Devinsky et al. and Wheless et al. suggested a linear relationship between exposure doses within the same dose range [52,156]. Wang et al. found that the time to peak plasma CBD concentrations and the mean acute elimination half-life were both shorter in the pediatric population than in adults [116]. Crockett et al. investigated the intra-individual variability of THC and suggested that it ranged from moderate to high (coefficient of variation [CV] 20–121%) [157]. No PK studies of THC in pediatrics have been reported.

Bansal et al. developed and verified a PBPK model to predict systemic CBD exposure in children aged 4–10 years, suggesting that pediatric patients have a comparable simulated CBD accumulation ratio to adults. In their models, the estimated fraction of CBD escaping the liver and plasma clearance in pediatrics was 155% and 63% of adults, respectively, and involving the ontogeny of UGT2B7 for children in the model made a minimal effect on the AUC and C_max_ of CBD [158]. Shenkoya et al. also simulated the dose of THC in breast milk and infant THC exposure with their paired lactation and infant THC PBPK model. They predicted that 0.34% to 0.88% of the mother’s THC dose was delivered to the infant, and the mother–infant plasma THC AUC_0–24h_ ratio increased to three times (3.4–3.6) with a six-fold increase in the amount of THC that was smoked by the mother [138].

#### 3.2.2. PK Profiles and Considerations in Pediatric Patients

The PK of pediatric patients could be complex due to the child’s growth and development process. No significant differences in CBD exposure between children and adolescents were observed in a clinical trial, while the CBD exposure in infants was also lower than in adolescents [52]. 

When administered orally, the first-pass metabolism of a drug is altered due to the immaturity of both metabolic enzymes and the gastrointestinal system, which could impact the oral bioavailability of drugs. For example, the total oral bioavailability of midazolam in preterm neonates is higher, approximately three times that of adults [159]. Gastric emptying differs in pediatric populations due to immaturity and does not approach the adult level until 6 to 8 months of age [160,161]. The influence of immature gastric emptying on the absorption rate has yielded conflicting results [161]. Pancreatic lipase secretion matures a few months after birth; therefore, the absorption of fat-soluble drugs is reduced in neonates and young children. In addition, the maturation of transporters should also be assessed when investigating DDIs for CBD/THC in pediatric populations. At birth, P-gp expression is not at adult levels in the intestine, liver, and blood–brain barrier [162,163].

The distributions of THC, CBD, and their metabolites are also different in pediatric populations. Generally, the fat content is relatively low in younger children [161,164]. The fat percentage of body mass peaks around six months of age and then drops to the level that it was at birth during the first ten years [164,165]. The high lipophilicity of THC makes it more sensitive to changes in fat percentage. Plasma protein binding matures in the early stages of life. For example, Maharaj et al. established a nonlinear model to describe the development of α-1-acid glycoprotein protein binding in pediatrics [166]. Additionally, the blood–brain barrier does not fully function after birth, which may result in a higher CBD/THC exposure in the brain [167].

Besides their absorption and distribution, the metabolism of THC, CBD, and their metabolites may also change in pediatrics. CYPs in the phase I metabolism of CBD and THC and UGTs in the phase II metabolism of CBD change rapidly after birth. For example, the CYP3A isoform activity varies in different development stages. The CYP3A4 activity in infants is higher than that in adults and then declines to the adult level. CYP3A4 protein increases rapidly after birth, from low levels to 50% of adult levels by 6 to 12 months of age [168]. At birth, the CYP2C9 activity levels are approximately 30% of the mature values. CYP2C9 protein increases rapidly from birth to five months of age, with half the population having CYP2C9 levels that are comparable to those of adults at five months. CYP2C9 concentrations and activity are relatively stable from 5 months to 18 years old [169]. The CYP2C19 activity is around 15% of the adult level at birth, increasing linearly in the first five months [169,170]. From five months to ten years, the CYP2C19 expression varied 21-fold, and the CYP2C19 concentration and activity level were comparable with adults after ten years of age. Like the phase I metabolism, the phase II metabolism changes during growth. It takes three months to three years or more to increase the activity of hepatic UGTs from <1.0% of adult levels in fetal to approach adult levels [161,162]. The metabolisms of 11-OH-THC, THC-COOH, CBD, 7-OH-CBD, and 7-COOH-CBD may vary at different stages of child growth due to the development of the UGT system, and the changes in the metabolism of the different compounds may also be varied at each growth stage.

#### 3.2.3. Drug Interactions and Their Impact on Pediatric Populations

Only a few studies have investigated DDIs in pediatric populations. Geffrey et al. found that when administered concomitantly with 25 mg/kg/day of CBD, clobazam concentrations increased by over 60%, and *N*-desmethylclobazam increased by 500% in 25 patients with refractory epilepsy aged 4–19 years [53]. Wheless et al. detected the DDI between CBD and clobazam in 61 pediatric patients aged 1 to 17 years. The CBD concentration increased by 2.5-fold as the victim. When CBD acted as the perpetrator, individuals taking a 40 mg/kg/day CBD oral solution exhibited 1.7- and 2.2-fold higher clobazam concentrations than those taking 10 mg/kg/day and 20 mg/kg/day cannabidiol oral solutions, respectively [52]. Gaston et al. discovered that CBD has DDIs when used concomitantly with topiramate, rufinamide, clobazam, and *N*-desmethylclobazam. Statistically significant (*p* < 0.01) increases in the concentrations of topiramate, rufinamide, and *N*-desmethylclobazam and a decrease in the concentrations of clobazam were observed with increasing the CBD dose from 5 to 50 mg/kg/day in 42 children. They also compared the differences in DDIs between adults and pediatrics, suggesting that DDIs with zonisamide and eslicarbazepine were only observed in adults [51]. A possible reason for this observation is that plasma zonisamide concentrations increase linearly with age, potentially influencing the DDI results [171].

Furthermore, Vaughn et al. used compartment PK models in the software to predict the DDI between serotonin reuptake inhibitors, escitalopram or sertraline, and cannabinoids in adolescents [172]. Their simulations suggested significant increases in both AUC_0–24h_ and C_max_ and increased risks of adverse effects after co-administration. When used with concurrent THC or low-dose CBD (5–15 mg/day), escitalopram’s AUC_0–24h_ and C_max_ increased by 35% and 25%, respectively, in simulation. Similarly, sertraline’s AUC_0–24h_ and C_max_ increased by 33% and 36%, respectively, in simulation. 

### 3.3. Older Adults

#### 3.3.1. Overview of Studies Exploring CBD/THC in Older Adult Populations

Researchers have conducted investigations into the treatment effects of CBD/THC for age-related diseases such as Alzheimer’s disease [173,174,175,176,177], dementia [178,179], and Parkinson’s disease [180,181,182,183,184]. However, most of these studies either focus on the PD of CBD/THC and the clinical endpoint of the disease or combine all age groups when analyzing the PK results. Although the importance of clinical trials in older adults has been recognized since 1979 [118,179,185,186,187], to the best of our knowledge, only Ahmed et al. investigated the PK of THC in older adults in two studies, one in healthy older adults and the other involving older adults with dementia [123,124]. In the randomized controlled trial with healthy older adults, highly variable plasma concentrations of THC, 11-OH-THC, and THC-COOH were reported. The THC AUC_0_–_2h_ (n = 11) ranged from 1.67 to 3.51 ng∙h/mL, and the C_max_ was 1.42 to 4.57 ng/mL for those within the two-hour sampling period (Table 3). The T_max_ of the THC of some older adults was not observed during the 120 min sampling periods, whereas the T_max_ of the younger adults mainly were within 60 min with the same THC dosage form. Notably, in this study, the final samples were collected 2 h post-dose. Due to the long half-life of THC, the PK profiles of THC were incomplete [123]. A similar delayed T_max_ (120–180 min) was also observed in trials with older patients with dementia. Additionally, the AUC_0_–_6h_ of older adults with dementia was two times higher than that of the younger adults, with a CV of up to 136% being observed (Table 3) [124]. It should be noted that in the study of older adults with dementia, THC was taken with food, while in the study of younger adults, THC was taken during fasting. Since food increases both the AUC and C_max_ of THC, the conclusion that AUC is increased in older adults with dementia may be biased.

#### 3.3.2. Age-Related Changes in PK and Implications for CBD/THC

Aging has an impact on every aspect of CBD/THC PK, from absorption to excretion. The stomach, intestine, and liver play the most important roles in the absorption of CBD and THC when they are administered orally; due to the high lipophilicity of CBD/THC, the pancreas may also influence their absorption. Lungs and the oral cavity can also affect absorption when inhaled, administered sublingually, or by means of an oromucosal spray. The possible characteristics of an aged stomach could be hypochlorhydria and a decreased volume of gastrointestinal fluids [188,189]. Although most in vivo studies suggested that the gastric emptying time increased in older adults [190,191,192], a conflicting study showed that the emptying rate of older adults remains within the same range as that of young adults [193]. The hypochlorhydria may decrease CBD/THC dissolution and thus change the systemic exposure, especially for the powder and capsules [194]. The volume of gastric and intestinal fluids decreases in older adults [195,196]. Since self-emulsifying drug delivery systems (SEDDSs) require water or gastrointestinal fluid for emulsification, their absorption may be delayed. In older adults, the blood flow in the intestine and the small bowel surface decreases, impairing absorption. The changes in the stomach and intestine may explain the delayed T_max_ observed in the two clinical trials of Ahmed et al. [123,124]. 

Due to its low solubility in water, CBD/THC is often given with oil or SEDDS to increase its bioavailability. Pancreatic lipases hydrolyze the lipids in the oil, and the SEDDS oil phases in the intestine [197], and a decrease in pancreatic lipase secretion due to senescence may eliminate the bioavailability advantage of oil formulations and SEDDSs [198]. For an inhalation formulation, the effect of aging on absorption varies. Although the overall lung function decreased in older adults, physiological changes such as an increased alveolar–capillary surface area, alveolar size, end-expiratory lung volume, and functional residual capacity, along with a reduced expiratory airflow, make the influence of the declined lung function on the trans-pulmonary absorption of CBD/THC uncertain [199,200,201]. Salivary gland secretion decreased with aging, and the quality of saliva changed, resulting in reduced absorption of sublingual formulations and oromucosal sprays through the oral cavity [202,203]. 

Changes in fat composition, cardiovascular senescence, and the decrease in plasma protein are the main factors affecting drug distribution during aging. Generally, the total fat mass increases in older adults, and the fat redistribution to peripheral tissues (e.g., abdominal fat) increases [204]. The volume of distributions of lipid-soluble CBD/THC and their metabolites may increase. Cardiovascular senescence could lead to a decline in cardiac output and heart failure [205], which mediates hypoperfusion in the peripheral tissues and delays the distribution of CBD/THC to the periphery [206]. Due to brain diseases in older adults, the permeability of the blood–brain barrier may increase, therefore increasing the distribution of CBD/THC in the brain [207]. Plasma protein, mainly lipoproteins, followed by albumin, is bound to 95–99% of plasma THC [104,105]. The change in lipoproteins and albumin may impact the concentration of CBD/THC. A 20% increase in lipoprotein levels was observed in older adults over 60 years old. Serum albumin levels decreased by 20% at age 70 [208], while there appears to be very little change in α1-acid glycoprotein (AAG) concentrations in older adults [209].

The liver impacts the metabolism of CBD and THC. Both the liver’s blood flow and volume decrease with aging. The liver may lose 20–40% of its volume and around 40–60% of its blood flow in older adults compared with younger adults [210,211]. In addition to the physiological change in the liver, senescence may affect the functions of drug-metabolizing enzymes [212]. Age-related changes in CYP enzymes have been assessed in various studies using both in vitro and in vivo methods and have yielded some conflicting results [213]. UGTs have not been reported to decrease with aging, but one animal study suggests that UGT1A1 decreases in elderly mice [213]. The liver also impacts absorption through the first-pass metabolism, an important factor in oral bioavailability. With the aforementioned changes in the liver and the high prevalence of fibrosis, the first-pass metabolism may be reduced, resulting in a higher AUC. 

THC is excreted mainly as acid metabolites, about 65–80% in feces and 20–35% in urine, and the amount THC being excreted as a parent drug is limited. The lower renal metabolite excretion can be explained by tubular reabsorption [4]. Like the liver, the kidney’s size, weight, and renal blood flow decreased in older adults. The kidney loses one-fourth of its weight in older adults, and the renal blood flow decreases by 10% per decade, starting at 40 years old. Moreover, the estimated glomerular filtration rate and renal tubular reabsorption capacity decreased [214]. The overall effect of aging on the renal excretion of CBD and THC and their metabolites is unknown. 

#### 3.3.3. Drug Interactions and Considerations for Older Adult Populations

In addition to the growing use of recreational cannabis in older adults, the potential medical use of cannabis by older adults is also on the rise for the treatment of pain, insomnia, anxiety, depression, and Parkinson’s disease [215,216]. On the other hand, polypharmacy is often encountered in older adults, which makes the DDI considerations particularly important in older adults, especially considering that CBD/THC could be either a perpetrator or victim in DDIs. However, there is a lack of data to substantiate either of these potential drug interactions in older adults. Some case reports investigated potential DDIs between CBD/THC and medications such as warfarin (85-year-old male) and clopidogrel that was co-administered with aspirin (76-year-old male) in older adults [71,97]. The international normalized ratio (INR) was unchanged in the case report of an 85-year-old male patient using warfarin. In contrast, five case reports in young adults revealed INR elevations in patients on warfarin receiving cannabis. However, the dose of CBD and THC was relatively low in the case report of the older adult, which was 5.3 mg CBD and 0.3 mg of THC once daily [97], compared with a daily dose of 10.3 mg CBD and 14.7 mg THC reported in the case report of DDI effects in young adults [98]. Brown et al. reported a case where the co-medication of CBD with clopidogrel and aspirin resulted in acute myocardial infarction in a 76-year-old man [71]. More clinical studies are essential to assess the DDIs between CBD/THC and other drugs in older adults.

### 3.4. Patients with Hepatic or Renal Impairment

#### 3.4.1. Overview of Studies Examining CBD/THC in Patients with Hepatic or Renal Impairment

PK studies of CBD/THC in patients with hepatic or renal impairment are limited. Interestingly, Suzuki et al. suggested that CBD demonstrated a treatment effect on renal fibrosis, significantly reduced apoptosis in the kidney, and partially protected the kidney from cisplatin-induced nephrotoxicity. The proposed mechanisms of the renoprotection effects are suppressing the mRNA of inflammatory cytokines in cisplatin-induced nephropathy and reducing apoptosis by inhibiting caspase-3 activity [217]. Another animal study showed that CBD exhibited significant renoprotection against acute kidney injury by favoring the development of regulatory T-17 (Treg 17) cells, thus inhibiting the proinflammatory responses that lead to kidney injury [218]. Two reviews also investigated the effect of CBD and CBD co-medicated with THC on symptoms associated with kidney failure, demonstrating positive effects on uremic pruritus, restless legs syndrome, anorexia, and nausea. These treatment effects may be due to the actions of CBD and THC on a range of neuronal receptors and ion channels, including CB1 receptors, CB2 receptors, gamma-aminobutyric acid A receptors, alpha-3 glycine receptors, 5HT1a receptors, transient receptor potential vanilloid type 1 (TRPV1), TRPV2, TRPV3, and TRPV4 [219,220]. For hepatic impairment, a study found that CB2 agonists reduced liver impairment and accelerated liver regeneration in mice [221].

#### 3.4.2. Impact of Hepatic Impairment on CBD/THC PK

Taylor et al. assessed the PK of a single oral dose of 200 mg CBD solution in healthy volunteers and patients with mild-to-severe hepatic impairment. The PK parameters of CBD in this trial are presented in Table 2. The results suggested that the AUC_0–t_ and C_max_ of CBD and its metabolites increase with the severity of hepatic impairment, except for 7-COOH-CBD in patients with severe hepatic impairment. The AUC_0–t_ and C_max_ were lower in patients with severe hepatic impairment than in healthy volunteers. The AUC_0–t_ of CBD was 44.3%, 134.7%, and 313.1% higher in patients with mild, moderate, and severe hepatic impairment, respectively, compared with volunteers with normal hepatic function. Similarly, the C_max_ of CBD was 57.4%, 139.2%, and 159.5% higher, respectively [117]. Bansal et al. developed and verified the effect of hepatic impairment on CBD by using PBPK modeling. In simulation, they captured the AUC and C_max_ change in patients with mild, moderate, and severe hepatic impairment [158]. To date, no clinical trials have reported the PK of THC in patients with hepatic impairment. Current prescribing information for Epidiolex^®^ recommends an initial dose adjustment and slower dose titration for patients with moderate-to-severe hepatic impairment [15].

#### 3.4.3. Impact of Renal Impairment on CBD/THC PK

Tayo et al. investigated the PK of a single dose of 200 mg CBD oral solution in subjects ranging from normal renal function to severe renal impairment, and the PK parameters of CBD in renal impairment patients are listed in Table 2. They suggested that the exposure to CBD and its major metabolites did not differ between healthy volunteers and patients with any degree of renal impairment. The AUC_0–t_ of CBD was 44%, 14%, and 15% higher in patients with mild, moderate, and severe renal impairment, respectively, compared with the normal renal function group. It is noted that the mild renal impairment group had the lowest mean body mass index, which may lead to an increase in CBD’s AUC_0–t_. According to their results, no dose adjustment of CBD is required in patients with renal impairment [118]. To date, no clinical trials have investigated the PK of THC in patients with renal impairment. 

#### 3.4.4. Drug Interactions and Considerations for Patients with Hepatic or Renal Impairment

Current research findings show no effect of renal impairment on the PK of CBD. Thus, drugs that affect renal function may have no impact on CBD, and no dose adjustment is needed for CBD in patients with renal impairment. To date, no clinical trials have reported the PK of THC in patients with hepatic or renal impairment. More clinical trials may be required to characterize the impact of DDIs in patients with hepatic and renal impairment. 

### 3.5. Other Special Populations

#### 3.5.1. CBD/THC Studies in Cancer Patients

Most of the CBD/THC studies in cancer patients focus on their PD effects, and only one pilot clinical study reported the PK of CBD and THC in patients with cancer. They found a large variability of both CBD and THC in cancer patients. The AUC_0–t_ of THC and CBD ranged from 1.11 to 6.61 ng∙h/mL and 0.49 to 4.06 ng∙h/mL, respectively, after a single dose of 2.5 mg THC and 2.5 mg CBD. Additionally, the t_1/2_ of CBD and THC increased with the dose. The median t_1/2_ for 2.5 mg of CBD was 0.75 h, while the median t_1/2_ for 7.5 mg of CBD was 1.53 h. Similarly, for 2.5 mg of THC, the t_1/2_ was 0.94 h, whereas for 7.5 mg of THC, the t_1/2_ was 1.39 h (Table 2 and Table 3) [119]. The effects of CBD/THC in cancer patients primarily focus on reducing symptom distress, including cancer-related pain, cachexia, nausea, and vomiting. However, conflicting results have been reported in clinical trials, and findings may lack statistical power [222,223,224,225]. A recent retrospective chart review suggested that there was no statistically significant change in symptom distress with medical cannabis use in cancer patients [226,227]. In contrast, some studies report that CBD/THC may be effective in cancer-related neuropathic pain and in pain that is inadequately controlled by opioid therapy [228]. Notably, a phase Ib trial showed that the CBD and THC complex has the potential to improve the one-year survival in glioblastoma patients, while no DDI effect of CBD and THC on temozolomide PK was observed [186].

#### 3.5.2. CBD/THC Studies in Patients with Neurological Disorders

Various clinical studies have been conducted with CBD as a potential treatment for neuropsychiatric disorders, including epilepsy, movement disorders, neuropathic pain, anxiety, schizophrenia, and substance abuse disorder; however, small sample sizes, conflicting results, and variability in study designs make determining its efficacy difficult [229,230,231,232,233]. The most rigorously tested and only FDA-approved indication for CBD is for the treatment of seizures associated with Lennox–Gastaut syndrome, Dravet syndrome, and tuberous sclerosis complex. Gherzi et al. reported the PK of CBD and THC in young patients with drug-resistant epilepsy [234]. Sativex^®^ has also been approved outside the US as an adjunctive treatment for spasticity related to multiple sclerosis [235]. A recent systematic review of randomized controlled trials comparing cannabinoids with conventional treatments or placebo found moderate evidence for analgesic benefit in patients with chronic neuropathic pain [236]. Additional studies have suggested that CBD may also be effective in treating symptoms of opioid addiction [237,238]. 

#### 3.5.3. CBD/THC Studies in Other Special Populations

In addition to the populations mentioned above, some research investigated the use of CBD/THC in several other populations. CBD and THC were reported to have the potential to benefit people with inflammatory bowel diseases [239,240]; the PK parameters of CBD and THC in Crohn’s disease are listed in Table 2 and Table 3. CBD also reduces pancreatic inflammation caused by immune cell invasion in mice, which can lead to type 1 diabetes [241]. On the other hand, a phase II study suggested that CBD has the potential to decrease the incidence of acute graft-versus-host disease [242]. Populations with HIV are another relative hotspot in CBD/THC studies. The effect of cannabis on the viral load in HIV-infected patients was investigated, and daily/near-daily cannabis use was associated with HIV viral load suppression [243,244]. Some studies also suggested that cannabis can improve pain and other medical symptoms of HIV [245,246,247,248]. Finally, cannabis was reported to enhance the global cognitive performance in older adult patients with HIV [249].

## 4. Conclusions

### 4.1. Summary of Key Findings on CBD/THC PK and Drug Interactions

The PK of CBD and THC have been characterized in several studies, but research gaps remain. Large inter-individual variability in PK has been observed in various publications, the reasons for which have not been well investigated. Due to the lack of research, there is also a knowledge gap about PK in special populations. DDIs between CBD/THC and a few therapeutic drugs, such as antiepileptic drugs, have been investigated in clinical trials, but more potential DDIs remain to be explored in clinical trials. As hemp-based CBD is legal at a federal level in the US and some CBD products contain THC, more special populations may be exposed to CBD and THC. Additionally, the absorption, distribution, and metabolism may differ in special populations, as mentioned in the section above. This emphasizes the necessity of evaluating potential DDIs between CBD/THC and medications that are commonly used in these populations.

### 4.2. Implications for Special Populations and Other Considerations

While well-designed clinical trials remain the gold standard for assessing PK and DDIs in special populations, challenges such as difficulty in recruitment, polypharmacy, individual genetic variation, and treatment compliance have hindered the conduct of such studies. The modeling approaches, particularly PBPK modeling, have become a method for predicting the PK in special populations and determine the potential DDIs. PBPK models incorporate systems biology to provide a more mechanistic approach for analyzing populations that are difficult to represent adequately. Some research groups have developed PBPK models for CBD or THC, as covered in the sections where the PK of CBD were investigated by modeling in special populations, including adults with hepatic impairment and children, while the PK of THC were also studied in pregnant and lactating women.

### 4.3. Future Research Directions and Areas of Further Investigation

However, PBPK relies on our knowledge of the target population’s physiology and the drug’s properties. The absorption of different CBD and THC formulations has not been well described by the established models yet. Moreover, the roles and contribution of CYPs and UGTs in the metabolism of CBD and THC and their metabolites vary widely among studies. In addition, there are conflicting findings regarding the physiology of special populations. Overall, more work is required to comprehensively investigate the PK and DDIs of CBD and THC in special populations. New in vitro and in vivo technologies, such as organ-on-a-chip, microsampling, and microdialysis, may also enhance our physiological understanding and improve model predictions. Utilizing these physiological models in combination with clinical trials and modeling approaches could help better manage DDIs associated with CBD and THC in special populations. 

## Figures and Tables

**Table 1 pharmaceutics-16-00484-t001:** Summary of cannabidiol (CBD) and delta-9-tetrahydrocannabinol (THC) metabolisms and these cannabinoids as perpetrators of DDIs.

Component	Cannabinoids	Key Points
Enzymes	CBD	Phase I (main): CYP3A4, CYP2C9, and CYP2C19 [36]; Phase II (minor): UGT1A9, UGT2B7, and UGT2B17 [38,41].
	THC	Phase I: CYP3A4, CYP2C9, and CYP2C19 [41].
Metabolites	CBD	Primary: 7-OH-CBD by CYP2C9 and CYP2C19; active [37,78]; Secondary: 7-COOH-CBD, enzyme unidentified; inactive [79].
	THC	Primary: 11-OH-THC by CYP2C9 and CYP2C19; active [41]; Secondary: 11-COOH-THC by CYP3A4, CYP2C9, CYP2C19, alcohol dehydrogenases, ALDHs, and aldehyde oxidase; inactive [41,79,83,84].
Perpetrator of drug interaction	CBD	Competitively inhibits CYP2B6 (K_i,u_: 3.4 μM), CYP2C9 (K_i,u_: 0.093–1.3 μM), CYP2C19 (K_i,u_: 0.050–1.1 μM), CYP2D6 (K_i,u_: 0.074 μM), CYP2E1 (K_i,u_: 0.021 μM), CYP3A4 (K_i,u_: 0.093 μM), and CES1 (K_i,u_: 0.091–0.97 μM) in vitro [41]; Time-dependently inhibits CYP1A1 (K_i,u_: 5.9 μM), CYP1A2 (K_i,u_: 0.11–0.59 μM), and CYP1B1 (K_i,u_: 3.1 μM) in vitro [41]; Mixed-type inhibits CYP1A2 (K_i,u_: 0.12 μM) in vitro [85]; Also inhibits UGT1A6 (IC_50,u_: 0.40 μM), UGT1A9 (IC_50,u_: 0.073–0.12 μM), UGT2B4 (IC_50,u_: 0.22 μM), and UGT2B7 (IC_50,u_: 0.82 μM) in vitro [41]; Induces CYP1A2, CYP2B6, and CYP3A4 in vitro, parameters not reported [86].
	THC	Competitively inhibits CYP1A2 (K_i,u_: 0.090 μM), CYP2B6 (K_i,u_: 0.25 μM), CYP2C9 (K_i,u_: 0.073 μM), CYP2D6 (K_i,u_: 0.11 μM), and CES1 (K_i,u_: 0.031–0.54 μM) in vitro [41]; Time-dependently inhibits CYP1A1 (K_i_: 1.2 μM) and CYP2A6 (K_i_: 0.86 μM) in vitro [41]; Mixed-type inhibits CYP2C19 (K_i,u_: 0.056 μM) in vitro [85]; Also inhibits UGT1A9 (IC_50,u_: 0.33–0.49 μM), UGT2B4 (IC_50,u_: 0.47 μM), and UGT2B7 (IC_50,u_: 1.4 μM) in vitro [41].
Additional Factors		Body weight; fat distribution/composition; protein binding

CYP, cytochromes P450; UGT, UDP-glucuronosyltransferases; ALDH, aldehyde dehydrogenase; CES1, carboxylesterase 1; DDI, drug–drug interaction; K_i,u_, apparent unbound inhibitory constant; IC_50,u_, 50% inhibitory unbound concentration; K_i_, apparent inhibitory constant.

**Table 2 pharmaceutics-16-00484-t002:** Pharmacokinetic parameters for cannabidiol (CBD) in special populations.

Population	Dose	AUC_0–t_ (ng·h/mL)	C_max_ (ng/mL)	t_1/2_ (h)	V/F (L)	Study
Pediatric patients with epilepsy	10 mg/kg/day, oral	173.9 (CV%: 103.3)	59.03 (CV%: 169.4)	31.3 (CV%: 74.9)	-	Wheless 2019 [52]
	20 mg/kg/day, oral	507.1 (CV%: 135.6)	110.5 (CV%: 128.8)	33.5 (CV%: 44.7)	-	Wheless 2019 [52]
	40 mg/kg/day, oral	914.5 (CV%: 126.3)	256.9 (CV%: 136.9)	21.6 (CV%: 48.7)	-	Wheless 2019 [52]
	12.2 mg/kg/day, oral	226.3 (70.5–861.3) ^1,2^	49.6 (14.4–302.0) ^1,2^	-	-	Guido 2021 [115]
	0.13–5.0 mg/kg, oral	-	13.1 (2.0–112.7) ^1^	6.2 (2.8–9.1)	2392 (188–9174) ^1^	Wang 2020 [116]
Patients with mild hepatic impairment	200 mg, oral	648 (44.2) ^3^	233 (70.5) ^3^	15.7 (58.3) ^3^	5302 (60.1) ^3^	Taylor 2019 [117]
Patients with moderate hepatic impairment	200 mg, oral	1054 (38.9) ^3^	354 (42.3) ^3^	20.5 (39.2) ^3^	4668 (40.1) ^3^	Taylor 2019 [117]
Patients with severe hepatic impairment	200 mg, oral	1855 (52.0) ^3^	381 (52.2) ^3^	22.1 (44.9) ^3^	2437 (70.5) ^3^	Taylor 2019 [117]
Patients with mild renal impairment	200 mg, oral	671 (40.9) ^3^	200 (42.7) ^3^	15.5 (CV%: 64.5)	6661 (CV%: 55.5)	Tayo 2020 [118]
Patients with moderate renal impairment	200 mg, oral	530 (74.4) ^3^	172 (85.3) ^3^	14.6 (CV%: 46.6)	7778 (CV%: 58.0)	Tayo 2020 [118]
Patients with severe renal impairment	200 mg, oral	532 (32.7) ^3^	155 (40.6) ^3^	13.1 (CV%: 41.5)	6016 (CV%: 39.9)	Tayo 2020 [118]
Patients with advanced cancer	2.5 mg, oromucosal spray	0.65 (0.49–4.06) ^1^	0.58 (0.48–2.45) ^1^	0.72 (0.57–0.86) ^1^	-	Clarke 2022 [119]
	7.5 mg, oromucosal spray	5.96 (1.51–12.15) ^1^	1.55 (0.62–2.25) ^1^	1.53 (1.16–7.06) ^1^	-	Clarke 2022 [119]
Adult patients with epilepsy	200–300 mg, oral	0.53 (0.26, (ng·h/mL)/mg) ^4^	0.03 (0.01, (ng/mL)/mg) ^4^	38.9 (19) ^4^	1515 (1024, L/kg) ^4^	Birnbaum 2019 [120]
Patients with multiple sclerosis	5 mg, oral	9.44 (CI: 8.5–10.4) ^5^	1.33 (CI: 1.20–1.46)	-	3912 (CI: 3565–4260)	Hansen 2024 [121]
	10 mg, oral	19.02 (CI: 17.0–21.0) ^5^	2.67 (CI: 2.39–2.95)	-	3781 (CI: 3341–4222)	Hansen 2024 [121]
	15 mg, oral	32.35 (CI: 31.6–33.1) ^5^	4.54 (CI: 4.44–4.65)	-	3435 (CI: 3345–3525)	Hansen 2024 [121]
Patients with Crohn’s disease	7.5 mg, oral	40.32 (SD: 25.65) ^5^	8.1 (SD: 5.4)	0.65 (SD: 0.42)	-	Naftali 2021 [122]

If not labeled, data are presented as mean (range). AUC_0–t_, area under the plasma concentration–time curve from time zero to last measurable concentration; AUC_0–inf_, area under the curve to infinity; C_max_, maximum measured plasma concentration; t_1/2_, terminal (elimination) half-life; V/F, apparent volume of distribution; CV, coefficient of variation; CI, 95% confidence interval; SD, standard deviation; -, data not reported. ^1^ Median (range). ^2^ At steady state. ^3^ Geometric mean (Geometric CV%). ^4^ Geometric mean (SD). ^5^ AUC_0–inf_.

**Table 3 pharmaceutics-16-00484-t003:** Pharmacokinetic parameters for delta-9-tetrahydrocannabinol (THC) in special populations.

Population	Dose	AUC_0–t_ (ng·h/mL)	C_max_ (ng/mL)	t_1/2_ (h)	V/F (L)	Study
Healthy older adults	3 mg	-	1.42 (0.53–3.48)	-	-	Ahmed 2014 [123]
	5 mg	-	3.15 (1.54–6.95)	-	-	Ahmed 2014 [123]
	6.5 mg	-	4.57 (2.11–8.65)	-	-	Ahmed 2014 [123]
Older adults with dementia	0.75 mg	0.88 (CV%: 124)	0.41 (CV%: 138)	5.08 (CV%: 39)	-	Ahmed 2015 [124]
	1.5 mg	2.01 (CV%: 136)	1.01 (CV%: 112)	5.06 (CV%: 37)	-	Ahmed 2015 [124]
Patients with advanced cancer	2.5 mg oromucosal spray	1.71 (1.11–6.61) ^1^	1.31 (0.76–2.94) ^1^	0.94 (0.75–1.14) ^1^	-	Clarke 2022 [119]
	7.5 mg oromucosal spray	8.26 (2.67–11.72) ^1^	2.35 (1.09–3.19) ^1^	1.39 (1.30–2.88) ^1^	-	Clarke 2022 [119]
Patients with multiple sclerosis	2.5 mg oral	3.23 (CI: 2.7–3.7)	0.57 (CI: 0.50–0.63)	-	3173 (CI: 2753–3594)	Hansen 2024 [121]
	5 mg oral	7.03 (CI: 6.6–7.5)	1.21 (CI: 1.15–1.26)	-	2906 (CI: 2728–3084)	Hansen 2024 [121]
	7.5 mg oral	10.81 (CI: 10.5–11.1)	1.85 (CI: 1.81–1.88)	-	2831 (CI: 2754–2909)	Hansen 2024 [121]
Patients with neuropathic pain	0.5 mg inhaled	5 (SD: 2.4) ^2^	14.3 (SD: 7.7)	-	-	Almog 2020 [125]
	1.0 mg inhaled	12.8 (SD: 5.5) ^2^	33.8 (SD: 25.7)	-	-	Almog 2020 [125]
	3.08 mg inhaled	10.1 (SD: 3.3)	38 (SD: 10)	-	-	Eisenberg 2014 [126]
Patients with Crohn’s disease	2 mg oral	10.7 (SD: 2.2)	3.0 (SD: 2.1)	0.55 (SD: 0.27)	-	Naftali 2021 [122]

If not labeled, data are presented as mean (range). AUC_0–t_, area under the plasma concentration–time curve from time zero to last measurable concentration; AUC_0–inf_, area under the curve to infinity; C_max_, maximum measured plasma concentration; t_1/2_, terminal (elimination) half-life; V/F, apparent volume of distribution; CV, coefficient of variation; CI, 95% confidence interval; SD, standard deviation; -, data not reported. ^1^ Median (range). ^2^ AUC_0–inf_.

## Data Availability

The data presented in this study are available in this article.

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
