# Peer review of "CBD and THC in Special Populations: Pharmacokinetics and Drug–Drug Interactions"

_pharmaceutics, 2024, doi:10.3390/pharmaceutics16040484_

Round 1

Reviewer 1 Report

Comments and Suggestions for Authors

This manuscript gave a very well organized, comprehensive review on drug-drug interaction caused by CDB and THC, with focus on special population. It is highly relevant for the use of these drugs and for further research regarding DDI caused by these drugs. A major issue in this review is the lack of a quantitative understanding of the DDIs caused by CDB and THC. It is not only necessary to know which enzymes or transporters could be involved in DDI, more relevant would be the understanding between the potency of these two drugs as inhibitors of these enzymes and transporters and their plasma exposure in the patients. These data are of special interest as the authors mentioned PBPK as future research direction in this field.

Author Response

We agree that a quantitative understanding of the DDIs caused by CBD and THC is essential for comprehensively describing this field. To address this, we have added Table 1, which summarizes the PK and DDI parameters of CBD and THC in special populations. Additionally, the quantitative unbound inhibition constant or 50% inhibitory unbound concentration of CBD and THC for different related enzymes has been included in Table 1 of the revised manuscript (Lines 164-168). Furthermore, we have made substantial changes to this manuscript to include more detailed information. 

Reviewer 2 Report

Comments and Suggestions for Authors

This manuscript by Qian et al. reviews the mostly unexplored topic of cannabinoid PK and DDI in special populations with remarkable thoroughness. Not only they provide a comprehensive overview of available literature through an impressive collection of 228 references, but wherever data are sparse they add their own informed speculations, which are amply supported by evidence. Therefore, this review is more than a mere compilation of existing (albeit often contradictory) clinical observations, as the authors integrate literature data with their own PK expertise to formulate predictions. As such, the manuscript is a highly valuable contribution to the field that identifies gaps in our knowledge and clearly sets out directions for future research. I have no major comments to add and recommend the manuscript to be accepted for publication in its current form.

Minor typos:

-          Line 399: Repetition of the word “function”

-          Line 503: “peripheral” should probably read “periphery”

-          Line 550, section title of 3.4.1.: First letter of “Renal” should be lowercase

-          Line 554: “renal” should probably be replaced by “kidney”

Line 581: Revise wording, “conducted” should probably be replaced by “investigated”

Author Response

We would like to thank the reviewers for your comments regarding our manuscript." Those comments are valuable and very helpful in improving our manuscript. We have carefully reviewed the comments and have modified the manuscript to reflect these changes. Revised portions are marked in red in the manuscript. Please find below the responses to the reviewer's comments.

  1. Reviewer comment: Line 399: Repetition of the word "function"

Response: Thank you for pointing this out. We've changed "Additionally, the function of blood-brain barrier does not fully function after birth and may result in a higher CBD/THC exposure in the brain." to "Additionally, the blood-brain barrier does not fully function after birth and may result in a higher CBD/THC exposure in the brain." (lines 442-444).

  1. Reviewer comment: Line 503: "peripheral" should probably read "periphery"

Response: Corrected. Please refer to line 547.

  1. Reviewer comment: Line 550, section title of 3.4.1.: First letter of "Renal" should be lowercase.

Response: Thank you for your comment. We have corrected the uppercase (line 594).

  1. Reviewer comment: Line 554: "renal" should probably be replaced by "kidney"

Response: Corrected. Please refer to line 598.

  1. Reviewer comment: Line 581: Revise wording, "conducted" should probably be replaced by "investigated"

Response: Thank you for bringing this to our attention. We have replaced the word. Please refer to line 639.

Reviewer 3 Report

Comments and Suggestions for Authors

The manuscript represents a good topic, but the following criteria need to be addressed:

1. The abstract is much generalized; it should be rewritten to indicate some specific information about the topic and a general conclusion. The abbreviations should be fully written for the first appearance.

2. The introduction section should contain detailed information about the reported pharmacokinetics of CBD and THC including absorption, distribution, elimination half-life and excretion in healthy volunteers.

3. Previous studies on pharmacokinetic parameters of CBD and THC in pregnant or lactating women should be mentioned and discussed.

4. ''The THC AUC0-2h (n=11) ranged from 1.67 to 3.51 ng/mL'' What did the authors mean by this sentence? The unit of AUC is not correct.

5. Schematic figures for the topic should be provided.

6. In tabulated form, comparisons of the pharmacokinetic parameters as K, t1/2, tmax, cmax, AUC, Vd and MRT between special populations could be provided.

7. More detailed discussion about the proposed mechanisms for CBD and THC renoprotection correlated with their pharmacokinetics is required.

8. More detailed discussion for the obtained pharmacokinetic profile of CBD in renal impairment is required.

9. More detailed information for the pharmacokinetics of CBD/THC in cancer patients should be provided.

10. Any data about the pharmacokinetics of CBD/THC in people with inflammatory bowel diseases, diabetes, HIV, epilepsy, movement disorders, neuropathic pain, anxiety or schizophrenia should be provided.

11. The English language should be accurately checked throughout the manuscript.

Comments on the Quality of English Language

The English language should be accurately checked throughout the manuscript.

Author Response

We would like to thank the reviewer for your comments regarding our manuscript." Those comments are valuable and very helpful in improving our manuscript. We have carefully reviewed the comments and have modified the manuscript to reflect these changes. Revised portions are marked in red in the manuscript. Please find below the responses to the reviewer's comments.

Reviewer 3:

  1. Reviewer comment: The abstract is much generalized; it should be rewritten to indicate some specific information about the topic and a general conclusion. The abbreviations should be fully written for the first appearance.

Response: Thank you for pointing this out. The abstract has been rewritten to include more specific information and a general conclusion. We also modified the abbreviations. Please refer to lines 13-27.

  1. Reviewer comment: The introduction should contain detailed information about the reported pharmacokinetics of CBD and THC, including absorption, distribution, elimination half-life, and excretion in healthy volunteers.

Response: Thank you for your comment. We appreciate your feedback and have revised the manuscript accordingly. Please refer to lines 81-87 for CBD and lines 89-95 for THC in the revised manuscript.

  1. Reviewer comment: Previous studies on pharmacokinetic parameters of CBD and THC in pregnant or lactating women should be mentioned and discussed.

Response: We tried to find plasma pharmacokinetic (PK) parameters in pregnant or lactating women; however, there was no related data. Therefore, we added "And there is a lack of cannabinoid plasma concentration-time data in pregnant or lactating women." in lines 322-323 and modified the sentence in line 347.

  1. Reviewer comment: ''The THC AUC0-2h (n=11) ranged from 1.67 to 3.51 ng/mL'' What did the authors mean by this sentence? The unit of AUC is not correct.

Response: Thank you for bringing this to our attention. We checked the reference and corrected the unit from "ng/mL" to "ng∙h/mL." Please refer to line 500. We apologize for the mistake.

  1. Reviewer comment: Schematic figures for the topic should be provided.

Response: We appreciate the reviewer's insightful suggestion and agree that incorporating schematic figures would enhance the clarity of our work. However, considering that comprehensive details were provided in the contents and three tables were newly included to elaborate on this topic, we believe that additional figures may be unnecessary.

  1. Reviewer comment: In tabulated form, comparisons of the pharmacokinetic parameters as K, t1/2, tmax, cmax, AUC, Vdand MRT between special populations could be provided.

Response: We would like to thank the reviewer for comment. The manuscript added two tables summarizing t1/2, Cmax, AUC, and Vd of CBD and THC, respectively, in different special populations. Please refer to lines 295-315.

  1. Reviewer comment: More detailed discussion about the proposed mechanisms for CBD and THC renoprotection correlated with their pharmacokinetics is required.

Response: We agree and have updated the section. Detailed proposed mechanisms for CBD and THC renoprotection have been added. Please refer to lines 597-604 and 607-612.

  1. Reviewer comment: More detailed discussion for the obtained pharmacokinetic profile of CBD in renal impairment is required.

Response: Thank you for the suggestion. We have added sentences: "The AUC0-t of CBD was 44%, 14%, and 15% higher in patients with mild, moderate, and severe renal impairment, respectively, compared with the normal renal function group. It is noted that the mild renal impairment group has the lowest mean body mass index, which may lead to an increase in CBD AUC0-t." (lines 634-638) as a detailed description and discussion. Also, the PK parameters in patients with renal impairment have been summarized in Table 2 (lines 295-305).

  1. Reviewer comment: More detailed information for the pharmacokinetics of CBD/THC in cancer patients should be provided.

Response: Thank you for bringing this to our attention. We have provided more detailed PK information of CBD/THC in cancer patients in lines 650-657. Also, the PK parameters in patients with cancer have been summarized in Table 2 (lines 295-305).

  1. Reviewer comment: Any data about the pharmacokinetics of CBD/THC in people with inflammatory bowel diseases, diabetes, HIV, epilepsy, movement disorders, neuropathic pain, anxiety or schizophrenia should be provided.

Response: The PK data of CBD/THC in inflammatory bowel diseases, epilepsy, movement disorders, and neuropathic pain have been summarized in Table 2 and Table 3. Please refer to lines 295-315.

  1. Reviewer comment: The English language should be accurately checked throughout the manuscript.

Response: Thank you for your comment. We have reviewed and carefully checked the English to ensure its clarity.

Reviewer 4 Report

Comments and Suggestions for Authors

I think the article is well written, it has novelty elements, relevant bibliographic notes. Please follow the few comments marked in the text.

Author Response

We would like to thank the reviewer for your comments regarding our manuscript." Those comments are valuable and very helpful in improving our manuscript. We have carefully reviewed the comments and have modified the manuscript to reflect these changes. Revised portions are marked in red in the manuscript. Please find below the responses to the reviewer's comments.

Reviewer 4:

  1. Reviewer comment: Line 121: Tiagabine is an antiepileptic drug.

Response: Thank you for pointing this out. The tiagabine has been moved to the antiepileptic drug group. Please refer to line 122.

  1. Reviewer comment: Line 121: Clozapine is not an antiepileptic drug.

Response: The clozapine has been moved to the non-antiepileptic drug group. Please refer to line 122.

  1. Reviewer comment: Line 183: "Bcrp" should be changed to "BCRP".

Response: Thank you for bringing this to our attention. We have changed the "Bcrp" to "BCRP". Please refer to line 186. We also changed the "Bcrp" to "BCRP" in line 158.

  1. Reviewer comment: Lines 259-260: Please fill in the phrase.

Response: Thank you for your comment. We have changed the "CYP2C9, CYP2C19, and CYP2D6 are responsible for 12.8%, 6.8%, and 20% of all clinical drugs." to "CYP2C9, CYP2C19, and CYP2D6 are involved in the metabolism of 12.8%, 6.8%, and 20% of the majority of clinical drugs, respectively." (lines 265-266). We hope this change makes it clear.

  1. Reviewer comment: Line 550: First letter of "Renal" should be lowercase.

Response: Thank you for pointing this out. We have corrected the uppercase (line 594).

  1. Reviewer comment: Lines 563-564: Exposure to CBD increases, but the concentration of metabolites is probably lower.

Response: Thank you for your valuable comment. We carefully checked the results in the reference paper. We found that for CBD and all its metabolites, only the AUC0-t and Cmax of 7-COOH-CBD in severe hepatic impairment patients were lower than those of the healthy population. We have modified this part to make it clear. Please refer to lines 615-619.

  1. Reviewer comment: Line 564: The AUC0-t is probably for CBD.

Response: Thank you for your valuable comment. We have corrected it. Please refer to line 619.

  1. Reviewer comment: Line 567: The Cmax is probably for CBD

Response: Thank you for your valuable comment. We have corrected it. Please refer to line 622.

Reviewer 5 Report

Comments and Suggestions for Authors

This article provides an overview of the pharmacokinetics and drug-drug interactions (DDIs) of CBD and THC in special populations such as pregnant and lactating women, paediatric populations, and older adult populations, etc…

The article reports an interesting topic such as the therapeutic potential of CBD and THC and the importance of understanding their pharmacokinetics (absorption, distribution, metabolism and excretion) and potential drug interactions in special populations to ensure safe and effective use. Although well written, there are still some minor issues, so modifications are needed.

1. I suggest to reorganise the introduction by making the text more concise, so that the purpose of the study is better understood.

2. How authors selected the articles chosen to review?. Which Database used? That keywords used? What are the criteria for including or excluding substances?

3. Some articles that should be added and discuss:

·         Alsherbiny MA, et al. Medicinal Cannabis-Potential Drug Interactions. Medicines (Basel). doi: 10.3390/medicines6010003

·         Monfort A, et al. Pharmacokinetics of Cannabis and Its Derivatives in Animals and Humans During Pregnancy and Breastfeeding. Front Pharmacol. doi: 10.3389/fphar.2022.919630.

·         Grant KS, et al. Cannabis use during pregnancy: Pharmacokinetics and effects on child development. Pharmacol Ther. doi: 10.1016/j.pharmthera.2017.08.014.

·         Carlier J, et al. Monitoring Perinatal Exposure to Cannabis and Synthetic Cannabinoids. Ther Drug Monit. doi: 10.1097/FTD.0000000000000667

·         Pérez-Acevedo AP, et al. Disposition of cannabinoids and their metabolites in serum, oral fluid, sweat patch and urine from healthy individuals treated with pharmaceutical preparations of medical cannabis. Phytother Res. doi: 10.1002/ptr.6931.

·         Busardò FP, et al. Disposition of Phytocannabinoids, Their Acidic Precursors and Their Metabolites in Biological Matrices of Healthy Individuals Treated with Vaporized Medical Cannabis. Pharmaceuticals (Basel). doi: 10.3390/ph14010059

·         Gherzi M, et al. Safety and pharmacokinetics of medical cannabis preparation in a monocentric series of young patients with drug resistant epilepsy. Complement Ther Med. doi: 10.1016/j.ctim.2020.102402.

·         Pichini S, et al. Δ9-Tetrahydrocannabinol and Cannabidiol Time Courses in the Sera of "Light Cannabis" Smokers: Discriminating Light Cannabis Use from Illegal and Medical Cannabis Use. Ther Drug Monit. doi: 10.1097/FTD.0000000000000683

4. I suggest shortening some paragraphs and using summary tables to make the text easier to read. For example paragraph 2.2. Mechanisms of CBD and THC interactions with other drugs This table is a suggestion and summarizes key points

Component

Key Points

Metabolic Enzymes

Both CBD and THC: Primarily metabolized by CYP3A4, CYP2C9, and CYP2C19.

CBD Metabolites

Primary: 7-OH-CBD, etc..

Inactive: 7-COOH-CBD

THC Metabolites

Main active: 11-OH-THC... further metabolized to

Inactive: e.g. 11-COOH-THC

CBD_DDI

Inhibits: CYP1A2, CYP2B6, CYP2C9, and so on… (in vitro).

Induces: CYP1A2, CYP2B6, and CYP3A4.

THC_ DDI

Inhibits: CYP1A2, CYP3A4, CYP2D6, and so on… (in vitro).

Clinical Examples (CBD)

Increased tacrolimus, clobazam, and … levels.

Clinical Examples (THC)

Increased buprenorphine, norbuprenorphine, and … levels.

Additional Factors

Lipophilicity

Protein binding

Another paragraph that can be shortened is “3.3.2. Age-related changes in PK and implications for CBD/THC”. I suggest a table like the following

Stage

Process

Effect of Aging

Absorption

Stomach & Intestine

Lungs

Oral Cavity

Distribution

Fat Composition

Cardiovascular Function

Blood-Brain Barrier

Plasma Proteins

Metabolism

Liver Function

Excretion

Kidneys

Comments on the Quality of English Language

none

Author Response

We would like to thank the reviewer for your comments regarding our manuscript." Those comments are valuable and very helpful in improving our manuscript. We have carefully reviewed the comments and have modified the manuscript to reflect these changes. Revised portions are marked in red in the manuscript. Please find below the responses to the reviewer's comments.

Reviewer 5:

  1. Reviewer comment: I suggest to reorganise the introduction by making the text more concise, so that the purpose of the study is better understood.

Response: Thank you for your valuable comment. We have reorganized the introduction. However, another reviewer suggested adding more detailed pharmacokinetics information to the introduction. We tried our best to balance the different comments and moved some parts of the introduction to later sections. We hope the revision helps you better understand our purpose.

  1. Reviewer comment: How authors selected the articles chosen to review?. Which Database used? That keywords used? What are the criteria for including or excluding substances?

Response: We agree with the reviewer that a detailed description of the article selection could be helpful. We add that information on lines 286-294. Please note that our manuscript is not a comprehensive review. We give an overview of the studies of CBD/THC and special populations and summarize our findings from those studies.

  1. Reviewer comment: Some articles that should be added and discuss:
  2. Alsherbiny MA, et al. Medicinal Cannabis-Potential Drug Interactions. Medicines (Basel). doi: 10.3390/medicines6010003
  3. Monfort A, et al. Pharmacokinetics of Cannabis and Its Derivatives in Animals and Humans During Pregnancy and Breastfeeding. Front Pharmacol. doi: 10.3389/fphar.2022.919630.
  4. Grant KS, et al. Cannabis use during pregnancy: Pharmacokinetics and effects on child development. Pharmacol Ther. doi: 10.1016/j.pharmthera.2017.08.014.
  5. Carlier J, et al. Monitoring Perinatal Exposure to Cannabis and Synthetic Cannabinoids. Ther Drug Monit. doi: 10.1097/FTD.0000000000000667
  6. Pérez-Acevedo AP, et al. Disposition of cannabinoids and their metabolites in serum, oral fluid, sweat patch and urine from healthy individuals treated with pharmaceutical preparations of medical cannabis. Phytother Res. doi: 10.1002/ptr.6931.
  7. Busardò FP, et al. Disposition of Phytocannabinoids, Their Acidic Precursors and Their Metabolites in Biological Matrices of Healthy Individuals Treated with Vaporized Medical Cannabis. Pharmaceuticals (Basel). doi: 10.3390/ph14010059
  8. Gherzi M, et al. Safety and pharmacokinetics of medical cannabis preparation in a monocentric series of young patients with drug resistant epilepsy. Complement Ther Med. doi: 10.1016/j.ctim.2020.102402.
  9. Pichini S, et al. Δ9-Tetrahydrocannabinol and Cannabidiol Time Courses in the Sera of "Light Cannabis" Smokers: Discriminating Light Cannabis Use from Illegal and Medical Cannabis Use. Ther Drug Monit. doi: 10.1097/FTD.0000000000000683

Response: We have included all the recommended references in the revised manuscript. Reference A by Alsherbiny MA, et al. has been discussed in lines 176-179; reference B by Monfort A, et al. has been discussed in lines 322-323; reference C by Grant KS, et al. has been discussed in lines 365-367; reference D by Carlier J, et al. has been discussed in lines 367-368; reference G by Gherzi M, et al. has been discussed in lines 674-675; and reference H by Pichini S, et al. has been discussed in lines 68-70.

  1. Reviewer comment: I suggest shortening some paragraphs and using summary tables to make the text easier to read. For example paragraph 2.2. Mechanisms of CBD and THC interactions with other drugs.

Response: We would like to thank the reviewer for the insightful comment. As suggested by the reviewer, we used a summary table (Table 1) to make section 2.2 easier to read. Please also refer to the revised manuscript's lines 145-146, 152-153, 164-168, 172-173, 180-181, 192-194, and 246-247.

  1. Reviewer comment: Another paragraph that can be shortened is "3.3.2. Age-related changes in PK and implications for CBD/THC".

Response: Thank you for your suggestion. We agree with your suggestion and attempted to condense the paragraph using a table. However, due to conflicting results requiring detailed discussion, we described them in sentences to ensure clarity. We anticipate that we may be able to summarize the age-related changes in the future when more studies become available.

Round 2

Reviewer 1 Report

Comments and Suggestions for Authors

The manuscript has been improved substantially and can be accepted in this form.

Reviewer 3 Report

Comments and Suggestions for Authors

The authors have made all the required corrections.

Reviewer 5 Report

Comments and Suggestions for Authors

The authors firstly answered reviewer'questions. Now the submission is acceptable for publication